# Prediction of Mortality in Older Hospitalized Patients after Discharge as Determined by Comprehensive Geriatric Assessment

**DOI:** 10.3390/ijerph19137768

**Published:** 2022-06-24

**Authors:** Chih-Hsuan Su, Shih-Yi Lin, Chia-Lin Lee, Chu-Sheng Lin, Pi-Shan Hsu, Yu-Shan Lee

**Affiliations:** 1Center for Geriatrics & Gerontology, Taichung Veterans General Hospital, Taichung 407219, Taiwan; inkflyhigh@gmail.com (C.-H.S.); sylin@vghtc.gov.tw (S.-Y.L.); billlin@vghtc.gov.tw (C.-S.L.); 2Department of Family Medicine, Taichung Veterans General Hospital, Taichung 407219, Taiwan; pshsu3@vghtc.gov.tw; 3Division of Endocrinology and Metabolism, Department of Internal Medicine, Taichung Veterans General Hospital, Taichung 407219, Taiwan; u502107@yahoo.com.tw; 4Institute of Clinical Medicine, College of Medicine, National Yang Ming Chiao Tung University, Taipei 112201, Taiwan; 5Department of Internal Medicine, Taichung Veterans General Hospital, Taichung 407219, Taiwan; 6Ph.D. Program in Translational Medicine, National Chung Hsing University, Taichung 402010, Taiwan; 7Department of Medical Research, Taichung Veterans General Hospital, Taichung 407219, Taiwan; 8Department of Post-Baccalaureate Medicine, College of Medicine, National Chung Hsing University, Taichung 402010, Taiwan; 9Graduate Institute of Microbiology and Public Health, College of Veterinary Medicine, National Chung Hsing University, Taichung 402010, Taiwan; 10Division of Neurology, Taichung Veterans General Hospital, Taichung 407219, Taiwan

**Keywords:** older people, comprehensive geriatric assessment, prediction model

## Abstract

Several dimensional impairments regarding Comprehensive Geriatric Assessment (CGA) have been shown to be associated with the prognosis of older patients. The purpose of this study is to investigate mortality prediction factors based upon clinical characteristics and test in CGA, and then subsequently develop a prediction model to classify both short- and long-term mortality risk in hospitalized older patients after discharge. A total of 1565 older patients with a median age of 81 years (74.0–86.0) were consecutively enrolled. The CGA, which included assessment of clinical, cognitive, functional, nutritional, and social parameters during hospitalization, as well as clinical information on each patient was recorded. Within the one-year follow up period, 110 patients (7.0%) had died. Using simple Cox regression analysis, it was shown that a patient’s Length of Stay (LOS), previous hospitalization history, admission Barthel Index (BI) score, Instrumental Activity of Daily Living (IADL) score, Mini Nutritional Assessment (MNA) score, and Charlson’s Comorbidity Index (CCI) score were all associated with one-year mortality after discharge. When these parameters were dichotomized, we discovered that those who were aged ≥90 years, had a LOS ≥ 12 days, an MNA score < 17, a CCI ≥ 2, and a previous admission history were all independently associated with one-year mortality using multiple cox regression analyses. By applying individual scores to these risk factors, the area under the receiver operating characteristics curve (AUC) was 0.691 with a cut-off value score ≧ 3 for one year mortality, 0.801 for within 30-day mortality, and 0.748 for within 90-day mortality. It is suggested that older hospitalized patients with varying risks of mortality may be stratified by a prediction model, with tailored planning being subsequently implemented.

## 1. Introduction

The aging of society is one of the most critical issues being faced this century. Due to a growing older population, enormous care burdens and expenses from hospitalizations being seen in this group cannot be overlooked [1]. In Taiwan, people over 65 years of age currently account for 15.7% of the population, with that number reaching more than 20% in the year 2025, and possibly 40% before the year 2060 [2]. It has been reported that medical expenditures on the older population account for 35.6% of the total medical expenditures in our country [3].

Older people admitted for acute care are at a high risk for lengthy hospital stays, more adverse events, higher readmission rates, and the need for long-term care. Additionally, hospital admission for the older patient is frequently associated with physical decline and mortality after discharge [4,5]. Previous research has reported that an increased age, functional disability, cognitive impairment at admission, multimorbidity, and poor nutritional status all predict a worsening of physical function after discharge [6], and later mortality [7], either in the short-term (e.g., within 1 or 3 months after discharge) [7,8] or long-term (beyond 3 months after discharge) [9,10]. Knowing these prognostic factors which surrounding different survival time periods are particularly important for old and frail patients who have approached the end of their lives, and this information will help towards providing individual care planning.

Comprehensive Geriatric Assessment (CGA) is a multidisciplinary process used in the evaluation and management of medical problems, physical function, psychosocial problems, and other issues [11]. A previous meta-analysis has reported that CGA is beneficial for improving survival in older people in both community and acute care settings [12]. It was found that older people tend to remain alive and in their own homes during follow-up if they had had a CGA performed during hospitalization [13]. Additionally, certain studies have shown that CGA is useful for predicting mortality rates in older patients living in acute care settings [14].

However, in Taiwan, use of the CGA to predict mortality in older people after hospitalization have been previously less frequently studied. In this study, incidences of mortality occurring within 1 year were examined in a group of hospitalized older patient, with death risk factors being examined through CGA. Based on the results, we intend to establish risk models for predicting short- and long-term all-cause death. It is hoped that these new prediction models can improve individualized clinical treatment and offer a better prognosis for older, hospitalized patients.

## 2. Materials and Methods

### 2.1. Study Participants

We retrospectively enrolled patients aged 65 years or older who were admitted from the emergency department to a geriatric ward during the period July 2012 to September 2017 at our hospital. Patients who expired during admission, could not complete a Comprehensive Geriatric Assessment, or had missing data were excluded. In the end, a total of 1565 patients were included. Because all data were based on patients being registered in the healthcare system’s Geriatric Assessment and Care Database of Taichung Veterans General Hospital, and subsequently analyzed anonymously in a retrospective manner, a verbal or written consent was not required from the enrolled subjects according to the regulations established by the ethics committee of the hospital. This study protocol was approved by the Institutional Review Board of Taichung Veterans General Hospital (IRB No: CE18256A).

### 2.2. Study Setting

The geriatric ward in Taichung Veterans General Hospital is a hospital facility devoted to care of those older patients who have been diagnosed with an acute illness at a tertiary teaching hospital in Taiwan. The staff of the geriatric ward consisted of four geriatricians, one nurse practitioner, and four case managers who were well-trained nurses. All nurse members who care for patients at the geriatric ward have received standard training in geriatric nursing skills for purposes of older patient care. Multidisciplinary team meetings are arranged once a week to report on the assessments of different specialists, treatment goals, complicated clinical problems, discharge plans and long-term care evaluation. Team members include geriatricians, rehabilitation specialists, dentists, physiotherapists, occupational therapists, dietitians, social workers, pharmacists, and psychologists.

### 2.3. Comprehensive Geriatric Assessment Evaluation

Comprehensive Geriatric Assessments is a multidimensional diagnostic process focusing upon medical, psychosocial, and functional problems, together with a coordinated management plan developed for treatment and follow-up. In our geriatric ward, CGAs are conducted by a well-trained nurse for all patients within 48 h of admission. Each patient’s basic demographic and medical information, including age, gender, Body Mass Index (BMI), Length of hospital Stay (LOS), fall episodes within one year, and number of admissions within 6 months were all collected. Polypharmacy was defined as the taking of 5 or more medications on a daily basis [15]. Basic Activities of Daily Living (ADL), both during baseline and after admission, were assessed using the Barthel Index (BI) [16]. The Instrumental Activities of Daily Living (IADL) both after admission and at baseline were evaluated through 8 complex activities: shopping, housekeeping, finance dealings, cooking, use of public transportation, telephoning, laundry, and the taking of medicine correctly [17]. A Mini-nutritional Assessment (MNA) was used to evaluate nutritional status [18]. The Charlson’s Comorbidity Index (CCI) was used for measuring a patient’s co-morbid condition [19].

### 2.4. Outcome

The primary outcome was mortality within one year after discharge.

### 2.5. Statistical Analysis

Continuous variables are expressed as median and interquartile range (IQR). Categorical data are expressed as numbers and percentage. The significance of the difference between groups was determined using the Mann–Whitney U test for continuous variables, and either the Chi square test or Fisher’s exact test for categorical variables. For examining factors associated with mortality, evaluation was performed using Cox regression analysis. In simple regression analysis, the independent variables included age, length of hospital admission, BI at baseline, BI at admission, IADL at baseline, IADL at admission, MNA, CCI, and number of admissions during the one year prior to admission. Factors with a *p* value of less than 0.05 identified from simple regression analysis were dichotomized for both multiple Cox regression analysis and the development of a scoring system for a prediction model based on adjusted HR in backward stepwise multiple Cox regression analysis. Cut-off values of dichotomization were according to a sensitivity test initially and adjusted by cut-off values published in the previous literature for extrapolation. The performance of the prediction model was tested using Receiver Operating Characteristic (ROC) analysis. The Youden Index was used to determine the optimal cut-off point [20]. All statistical analyses were performed using IBM SPSS version 22 (SPSS Inc., Chicago, IL, USA). For all tests, a *p* value (two-tailed) of less than 0.05 was considered statistically significant.

## 3. Results

A total of 1565 older patients (median age 81 years, IQR:74.0–86.0; 61.7% male) were enrolled (Table 1). Median length of stay in the geriatric ward was 9 days (IQR:6–14). Forty-point eighty-three percent (40.83%) of the older patients had fallen at least once in the year prior to admission. Polypharmacy was evident in 1012 (64.7%) patients. Median CCI was 2 (IQR:1.0–3.0). Overall, 110 older patients (7.0%) were confirmed dead one year after discharge from the geriatric ward. The patients who died had lower BI, IADL, and MNA scores at admission, as well as higher polypharmacy numbers, morbidities, previous admission numbers, and longer LOS (Table 1).

In order to establish a risk model for predicting all-cause death, we analyzed the risk factors affecting mortality within one year after discharge. There, it was established that age, LOS, polypharmacy, BI and IADL at both baseline and admission, MNA scores, CCI, and number of admissions were all predictors in simple Cox regression analysis (Table 2). When these parameters were dichotomized according to cut-off values published in previous literature [18,21,22,23], it was shown that age ≥ 90, LOS ≥ 12, polypharmacy, IADL at baseline > 0, IADL at admission > 0, MNA < 17, CCI ≥ 2, and number of admissions >0 were also found to be associated with mortality within one year. With these risk factors identified from CGA, we intended to develop the prediction mortality model accordingly. It was shown that in backward stepwise multiple Cox regression analysis, age ≥ 90, LOS ≥ 12, MNA < 17, CCI ≥ 2, and previous admissions were all independently associated with an increased risk of one-year mortality. Based on adjusted HR in multiple Cox regression analysis, the individual score of each risk factor was identified from a calculation of total risk score as follows: 1 point was added to those individuals aged ≥ 90, having a length of stay ≥ 12 or an MNA < 17, while 2 points were added to those with a CCI ≥ 2 or previous hospital admission within 6 months ≥ 1, with a possible maximal risk score of 7 points. The relationship between one-year mortality rates and summation risk scores is presented in Figure 1, with a progressive higher one-year mortality rate being observed when there is an increase in the summation risk score. To identify the cut-off value for the risk score, we utilized ROC analysis and calculated Youden index (Table 3). It was shown that a cut-off point ≥ 3, the area under curve (AUC), was optimal for predicting one-year mortality with a value of 0.691 (95% CI 0.642–0.740), with sensitivity and specificity being 79.1% and 53.5%, respectively, according to the best Youden index. Furthermore, we also determined the prediction for 30- and 90-day mortality. For risk summation scores ≥ 3, we found that the AUC was 0.801 and 0.748, respectively, for 30- and 90-day mortality, and the Youden index was 0.346 and 0.455, respectively.

## 4. Discussion

This retrospective study was aimed at identifying mortality risk factors in older patients within one year after discharge from a geriatric ward. Additionally, a prediction model of one-year mortality was also developed based on these risk factors, including age, length of stay, malnutrition, multi-morbidities, and previous admission history. In previous reports, mortality rates were found to be approximately 6% to 22% in older hospitalized patients one year after discharge [24,25]. In our study, the mortality rate was 7.7%, which was approximately equal to the result seen in previous studies. Regarding our patients, the average length of hospital stay was approximately 2 weeks, with a high proportion of these older patients having an issue with polypharmacy. Additionally, our patients experienced a decline in physical function as determined by the Barthel Index, as well as experiencing comorbidities.

In our study, it was shown that an older age, a longer length of hospital stay, previous admission history, polypharmacy, decline in physical function (as seen in ADL and IADL scores), malnutrition (as determined by a higher MNA score), and morbidity numbers were all found to be prediction factors for mortality, a finding which corresponds with previous studies [7,26]. Although in this study, dependency on ADL and IADL scores was significantly related to mortality statistically only in univariate analysis, the decline in ADL scores cannot be overlooked because functional dependence is consistently reported as being related to death in hospitalized older patients [27]. It remains important for hospital medical staff to pay more attention to changes in the ADL scores of older hospitalized patients. In addition to causing disability, previous research has also shown that in-hospital mortality or mortality after discharge is related to morbidity [28]. In our research, both morbidity and disability played key roles in mortality after discharge, which was in line with previous reports from another hospitalist ward, although the causal relationship between multimorbidities, physical functional decline, and mortality still requires further development.

In our study, it was found that those of an older age, particularly the oldest group (age ≥ 90), an increased length of stay upon one’s index admission, as well as a patient’s previous hospitalization history, were each associated with mortality occurring within one year, a finding compatible with a previous report [29]. In a large national study previously performed in France, the overall survival rate of patients admitted to an ICU was approximately 60% at 3 years. Mortality rates increased in proportion to age, particularly in those 80 years and older [30]. After an acute event, the role of aging facilitates the process of organ failure, and as a result, any acute illness deemed less severe may cause a subsequently poor prognosis. These results suggest that post discharge strategies may need to be adjusted according to different ages. Previous studies have also reported that longer lengths of stay and recurrent admissions were associated with post-discharge mortality in hospitalized stroke, chronic obstructive pulmonary disease, and pneumonia-acquired patients [31,32,33]. An extended LOS and recurrent hospitalization, either due to repeat acute illness or associated medical complications such as nosocomial infections and thromboembolic disease, may further predispose decompensation of underlying diseases, and thus directly and indirectly hinder clinical outcomes, including increased mortality. Therefore, prolonged stays and a repeat admission history may indicate an underlying frailty, thus requiring more careful attention being given by medical staff.

Malnutrition and its negative health outcomes are of great relevance in older people, as the condition predicts hospitalization, infectious diseases, and death [34]. The current study confirmed previous results from other surveys which used various malnutrition screening tools, including the Subjective Global Assessment (SGA) [35] and Malnutrition Universal Screening Tool (MUST) [36], while expanding on similar data generated using the Mini Nutritional Assessment, tool [37] as well as ours. Generally, some 20% to 50% of patients are undernourished prior to being hospitalized [34]. In our study, 25% older patients were malnourished and 37% were at risk of malnutrition, according to MNA. In view of high prevalence of nutritional impairment in our patients, and adverse effect on outcomes, it is important to have an early identification and appropriate management of this clinical condition which could make for a better prognosis of malnourished patients, while also cutting down on both the length of hospitalization stays and the medical costs attributed to hospitalization [38].

Comprehensive Geriatric Assessment (CGA) is a multidimensional diagnostic process focusing on medical, psychosocial, and functional problems. Importantly, these main components of CGA can be translated into a risk score to predict short- and long-term mortality in several independent cohorts of hospitalized patients. Based on CGA, this study identified that age, length of hospital stay, polypharmacy, physical function, malnutrition, multi-morbidities, and number of previous admissions were all related to one-year mortality. Taking into account these relevant factors, we established a prediction model to determine all-cause death within 30 days, 90 days, and 1-year according to the independent predictive effect of age, length of stay, malnutrition, multi-morbidities, and previous admission history, with a moderate to good predictive ability. It is implied that information collected and calculated from CGA can help stratify hospitalized older patients into groups that are at varying risks of mortality. Although prediction accuracy for one-year mortality was acceptable, it improved when predicting short-term mortality. In older patients, a risk score ≥ 3 could better predict short-term mortality within 30 and 90 days. One explanation for the improved discrimination may be due to those mortality predictors which play more important roles in the short-term death of older patients after discharge. Previously, two risk scoring systems, the RESPECT [39] and CHESS scales [40], had been established for older people in a community setting with resulting AUCs of approximately 0.75 and 0.66, respectively. However, these two prediction models were not developed for older patients being treated in an acute care unit. Our model still needs to be validated in order to provide better outcome prediction accuracy in older hospitalized patients after discharge. Nevertheless, in view of a potentially high mortality rate observed in older frail patients after acute hospitalization in Taiwan [41], it is proposed that our easy-to-use predictive tool for mortality after discharge can still be valuable for these older patients in the country. Through mortality risk stratification, it is feasible to identify high-risk populations, which may better help guide medical staff towards making individual plans for patients after bridging them from hospitalization to outpatient care. Consequently, appropriate intervention may be designed specifically for each individual.

There were some limitations to our study. First, this study was retrospective in design and could only show the association between predictors and outcomes. Additionally, only all-cause mortality was evaluated, and exact cause of death was not assessed. Third, the included patients came from a single hospital, most of whom were admitted from the emergency department. This may have limited the applicability of the results. Finally, this study was limited, in that we did not test our prediction model in another population for purposes of validation. Additionally, which subset of patients for whom our prediction model could be of particular use to was not examined. For example, a higher frequency of multi-component CGA abnormalities were more likely to occur in frail individuals, so these particular patients may have benefited from this prediction model much more. Overall, further developments such as multiple center prospective studies may be considered with intervention like early rehabilitation and program, improving physical function, and sequential evaluation including at admission, during intervention, before discharge, and after discharge. These remain necessary in order to better elucidate the limitations.

## 5. Conclusions

In this study, we discovered that older patients experiencing a high prevalence of polypharmacy, prolonged admission time, and multiple comorbidities are being admitted to geriatric wards due to acute illness. We also determined that hospitalized older patient with prolonged hospitalization time, poor nutrition, multiple comorbidities, and frequent hospitalizations were at high risk of mortality in the near future. As these risk factors are often associated with frailty in older people, CGA services are available to help identify and examine various risk domains, and subsequently a prediction model could therefore be implemented to identify a high-risk population, particularly in frail and older adults. Afterwards, tailored intervention involving proper medical care during the end-of-life cycle may ultimately be applied.

## Figures and Tables

**Figure 1 ijerph-19-07768-f001:**
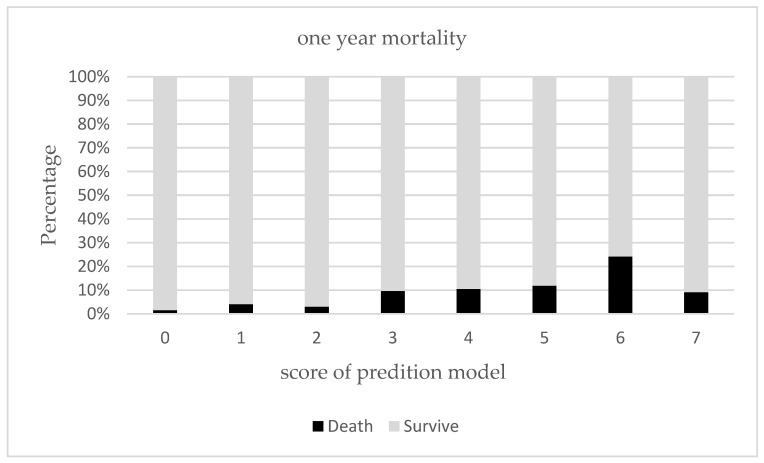
Prediction of one-year mortality using prediction model.

**Table 1 ijerph-19-07768-t001:** Basic characteristics.

	Total Patients (*N* = 1565)		Alive Patients (*n* = 1455)		Deceased Patients (*n* = 110)		*p* Value ^e^
Age (years)	81	(74.0–86.0)	81	(74.0–86.0)	83	(76–87)	0.075
Gender, Male (%)	966	(61.7%)	896	(61.6%)	70	(63.6%)	0.669
Length of stay (Days)	9.0	(6–14)	9	(6–14)	12	(7–17)	0.042
Falls in one year (%)	639	(40.8%)	589	(40.5%)	50	(45.5%)	0.306
Polypharmacy (%)	1012	(64.7%)	930	(63.9%)	82	(74.5%)	0.025
BI at baseline ^a^	90	(60–100)	90	(60–100)	80	(45–95)	0.001
BI at admission ^a^	55	(20–80)	55	(20–80)	45	(10–70)	0.004
IADL at baseline ^b^	4	(1–6)	4	(1–6)	2.5	(0–5)	<0.001
IADL at admission ^b^	2	(0–4)	2	(0–4)	1	(0–3)	0.004
MNA ^c^	21.5	(17.5–24.5)	22	(17.5–25.0)	18.7	(15–22)	<0.001
CCI ^d^	2	(1.0–3.0)	2	(1.0–3.0)	3	(2.0–4.0)	<0.001
Number of admissions	0	(0–0)	0	(0–0)	0	(0–1.0)	<0.001
Mortality	110	(7.0%)	--	--	--	--	--

Continuous data are expressed as median (IQR), Categorical data are expressed in number and percentage. ^a^ BI = Barthel Index. ^b^ IADL = Instrumental activities of daily living. ^c^ MNA = Mini-nutritional assessment. ^d^ CCI = Charlson comorbidity index. ^e^ Mann–Whitney U test for continuous variables and Chi square test or Fisher’s exact test for categorical variables.

**Table 2 ijerph-19-07768-t002:** Risk factors and Cox regression analysis (one year mortality).

	Simple Cox Regression Analysis (Original Data)			Simple Cox Regression Analysis (Dichotomized *)			Multiple Cox Regression Model			RISK SCORE
Variables	HR	95% CI	*p* value	HR	95% CI	*p* value	Adjusted HR	95% CI	*p* value	--
Age	1.022	0.998–1.047	0.066	1.840	1.133–2.986	0.014 **	1.750	1.074–2.853	0.025 **	1
Gender (male vs. female)	0.917	0.622–1.352	0.662	0.917	0.622–1.352	0.662	--	--	--	--
Length of stay (days)	1.018	1.003–1.032	0.016 **	1.802	1.240–2.619	0.002 **	1.551	1.058–2.275	0.024 **	1
Polypharmacy	1.644	1.070–2.525	0.023 **	1.644	1.070–2.525	0.023 **	--	--	--	--
BI at baseline	0.993	0.988–0.999	0.016 **	0.925	0.519–1.651	0.793	--	--	--	--
BI at admission	0.992	0.986–0.998	0.005 **	0.785	0.524–1.176	0.241	--	--	--	--
IADL at baseline	0.887	0.829–0.949	<0.001 **	0.616	0.408–0.930	0.021 **	--	--	--	--
IADL at admission	0.882	0.811–0.960	0.003 **	0.657	0.448–0.964	0.032 **	--	--	--	--
MNA	0.913	0.881–0.945	<0.001 **	2.141	1.449–3.165	0.001 **	1.508	1.002–2.269	0.049 **	1
CCI	1.319	1.190–1.461	<0.001 **	2.540	1.450–4.451	0.001 **	2.173	1.234–3.825	0.007 **	2
Previous admission history	1.642	1.419–1.899	<0.001 **	2.850	1.951–4.162	<0.001 **	2.418	1.644–3.557	<0.001 **	2

BI = Barthel Index. IADL = Instrumental activities of daily living. MNA = Mini-nutritional assessment. CCI = Charlson comorbidity index. * Dichotomized with AGE ≥ 90, Length of stay ≥ 12, BI ≥ 20, IADL ≥ 0, MNA < 17, CCI ≥ 2, Number of admissions > 0. ** *p* value < 0.05.

**Table 3 ijerph-19-07768-t003:** Sensitivity, specificity, and Youden index at different cut-off points of prediction model.

Cutoff Value	Sensitivity	Specificity	Youden Index	AUC	CI
Model for 1-year mortality	--	--	--	0.691	0.642–0.740
≥1	0.973	0.129	0.102	--	--
≥2	0.927	0.208	0.135	--	--
≥3	0.791	0.535	0.326 *	--	--
≥4	0.509	0.736	0.245	--	--
≥5	0.273	0.889	0.162	--	--
≥6	0.145	0.961	0.106	--	--
≥7	0.009	0.993	0.002	--	--
Model for 30 days mortality	--	--	--	0.801	0.711–0.891
≥1	1.000	0.123	0.123		
≥2	1.000	0.201	0.201	--	--
≥3	0.938	0.517	0.455 *	--	--
≥4	0.688	0.723	0.411	--	--
≥5	0.500	0.882	0.382	--	--
≥6	0.250	0.955	0.205	--	--
≥7	0.000	0.993	−0.007	--	--
Model for 90 days mortality	--	--	--	0.748	0.681–0.814
≥1	1.000	0.125	0.125	--	--
≥2	0.956	0.204	0.160	--	--
≥3	0.867	0.524	0.391 *	--	--
≥4	0.622	0.729	0.351	--	--
≥5	0.378	0.886	0.264	--	--
≥6	0.178	0.957	0.135	--	--
≥7	0.000	0.993	−0.007	--	--

* The largest Youden Index.

## Data Availability

Not applicable.

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
