# Peer review of "Prediction of Mortality in Older Hospitalized Patients after Discharge as Determined by Comprehensive Geriatric Assessment"

_ijerph, 2022, doi:10.3390/ijerph19137768_

Round 1

Reviewer 1 Report

Page 3.

Please check for minor typing errors (see page 3,  2nd paragraph, for example) and consistently report in full the acronyms ( see, for instance, IQR).

Page 4.

Line 2.  “.. within one-year mortality risk factors after discharge”  Please  anticipate and make here clearer the distance you considered after discharge

Line 4 from the bottom of the page. A total risk score is described. Please clearly mention the foundation you’re your choice. This is also relevant also because in the discussion (page 7) you contrast your choice with other methods.

Page 6

End first paragraph. Here and on another occasion in the paper Authors mention the need for further research. I suggest authors make explicit what they refer to or they may group all these “needs”  as “further developments” in the last paragraph of the discussion where they actually focus on the limits of the study.   

Page 7 first paragraph.   Here  authors may  comment more on the impact of the choices they made about the risk scoring system 

My main comment refers to the description and representation of results. A graphic representation of the most important results that the authors point out after the analysis would help the reader understand the potential applied impact of the study.  And in developing this representation, the limits they mention can turn on positive elements used to develop a graphical model reader can find useful in  the specific conditions authors refer to. 

I also suggest taking the concept of "frailty" in the conclusion. 

Reviewer 2 Report

Accept paper with one comment noted below.

I suggest abbreviations used in tables be written out at the bottom of the tables.  This would make the tables much easier to read. 

I suggest the paper be accepted with my only comment noted.  Well prepared study. 

The English is acceptable with no adjustment needed.

Reviewer 3 Report

Thank you for the possibility to read this paper. It is important to always strive to improve the care for frail older patients and to identify those with special needs. The overall impression is that the identified risk factors are all relevant and confirm what most clinicians know already. The work with establishing an index is not sufficient and it is doubtful if the index is relevant to clinical work. 

Language: 

The language has to improve. Many typos/misspellings, long sentences, wrong time/number etc. The text does not read easy. 

E.g. typo in line 51 ... are aT high risk of lengthy...

The language needs to be more precise and concise to ease the reading. E.g. not concise in line 64 "... and other issues." - which issues?  

Topic: 

CGA is not only an evaluation of health status - it is also planning of treatment/interventions. In this paper it is stated in lines 102-103 that CGA was performed by a well-trained case manager. The CGA cannot alone be performed by a case-manager - but this person can of course perform the tests/assessents. 

It is not really described how CGA consists of domains and that these domains can be assessed by the use of tests and scores. CGA is multidisciplinary assessment and will always need a physician/nurse involvement and evaluation of the tests and general impression to make a prioritized list of problems to solve. 

The chosen domains/tests are all appropriate for the assessment. However, it would improve the overall impression if the chosen cut-offs in dichotomized variables were made from sensitivity tests and not only based on previous literature (line 153-154). It cannot be of any surprise to anyone that you are in high risk of death after hospitalization if you are >90 years of age. It would be more interesting if it was assessed in more detail and based on the data. Also, regarding the chosen risk factors - it is not clear how long back the history of previous admission goes? Is it within 3 months? 1 year? Ever? And when is it interesting and possible a clinical meaningful variable to use? 

The index building lines163- It is not clear how the points are chosen. It is stated that it is due to HR(s) in the multiple Cox regression analysis - but the it is not clear why age >90 years with HR 1,75 only gives 1 point whereas LoS >12 days and MNA<17 gives 2 points with HR 1,5... 

The overall ability of the index (cut-off 3 points) to identify risk of 30+90 days risk of death is high but the sensitivity is 0,5. It would be a strength to the study if hte index was validated in a new population. The data in this paper are getting old - the world has changed and probably also treatment of acutely ill elderly. 

The question is whether this index is of any clinical importance if it should be used to guide clinicians to better care when it only can tell you that an acutely admitted person >90 with LoS >12 days (very long), with poor nutritional status and high comorbidity is of high risk of death. It does not add big value to common clinical knowledge. 

The discussion is too long and poorly distributed. It needs to be more concise and precise. 

References: Extremely many references. 66 references cannot all be important. Must be evaluated to include only relevant literature - e.g. more reviews to cover the different topics. It seems as if the journal has no upper limit of references? 

Conclusion: Lines 282-284 -  one does not know how long the LoS is before it has happened and it seems backwards to conclude that elderly patients with long LoS are admitted to a geriatric ward due to long LoS. It can be doubted that introduction of this Index would be useful in the clinic or if it would just add extra administrative work. However, the CGA and testing is of big value to all frail older patients and can be used when planning tailored interventions and care and identification of specific risk factors such as high age, malnutrition, multimorbidity, and history of admissions can be helpful to the clinician when communication with patients and relatives about the expectations for the admission and lifespan. 

Round 2

Reviewer 3 Report

Thank you for the possibility to read through the revised paper. I find that the paper has improved significantly and that the authors have addressed the issues raised in the review. 

I will look forward to see the validation of the index in a geriatric population and learn how it will be used to improve the course of hospitalization for frail older patients with acute care needs.